# Quantum Chaos, Random Matrices, and Irreversibility in Interacting Many-Body Quantum Systems

**DOI:** 10.3390/e24070959

**Published:** 2022-07-11

**Authors:** Hans A. Weidenmüller

**Affiliations:** Max-Planck-Institut für Kernphysik, D-69029 Heidelberg, Germany; haw@mpi-hd.mpg.de

**Keywords:** master equation, equilibration, Markov approximation

## Abstract

The Pauli master equation describes the statistical equilibration of a closed quantum system. Simplifying and generalizing an approach developed in two previous papers, we present a derivation of that equation using concepts developed in quantum chaos and random-matrix theory. We assume that the system consists of subsystems with strong internal mixing. We can then model the system as an ensemble of random matrices. Equilibration results from averaging over the ensemble. The direction of the arrow of time is determined by an (ever-so-small) coupling to the outside world. The master equation holds for sufficiently large times if the average level densities in all subsystems are sufficiently smooth. These conditions are quantified in the text, and leading-order correction terms are given.

## 1. Introduction

We investigate the connection between the irreversible statistical equilibration of closed quantum systems, quantum chaos [1], and random-matrix theory [2]. We do so by expanding and generalizing ideas formulated in Refs. [3,4].

To begin, we recall the standard theoretical description of statistical equilibration. The interaction between the constituents of an isolated (or closed) many-body quantum system drives the system toward statistical equilibrium. The process is theoretically most simply described in terms of the Pauli master equation. Given a division of the system into a finite number *K* of subsystems, the master equation for the time-dependent occupation probability $P_{E,\alpha }\left(t\right)$ of subsystem α at excitation energy *E* reads
(1)$$\begin{array}{c} {\dot{P}}_{E,\alpha }\left(t\right)=\sum_{\beta }\left(R_{\beta \to \alpha }P_{E,\beta }\left(t\right)-R_{\alpha \to \beta }P_{E,\alpha }\left(t\right)\right). \end{array}$$
The dot indicates the time derivative, and the indices α,β range from 1 to *K*. Equation (Equation 1) conserves probability, $\sum_{\alpha }{\dot{P}}_{E,\alpha }=0$. The interaction between states in subsystems α and β gives rise to the time-independent transition rates $R_{\alpha \to \beta }$ and $R_{\beta \to \alpha }$. These obey detailed balance,
(2)$$\begin{array}{c} \rho_{E,\beta }R_{\beta \to \alpha }=\rho_{E,\alpha }R_{\alpha \to \beta }. \end{array}$$
Here, $\rho_{E,\alpha }$ is the average level density of the states in subsystem α at excitation energy *E*. Equation (Equation 2) implies that $P_{E,\alpha }^{\left(0\right)}=C\rho_{E,\alpha }$ with *C* a constant independent of α is a time-independent solution of Equation (Equation 1). That solution describes statistical equilibrium. For large times, the solutions $P_{E,\alpha }\left(t\right)$ of Equation (Equation 1) tend generically toward the equilibrium solution $P_{E,\alpha }^{\left(0\right)}$, irrespective of the initial conditions at time t=0. Thus, Equation (Equation 1) describes the irreversible development in time of the system toward statistical equilibrium.

Equation (Equation 1) goes back to the early days of quantum theory. The master equation is a standard topic in textbooks on quantum statistical mechanics, and it is widely used to describe statistical equilibration in atomic, molecular, and nuclear systems. At the same time, equilibration in isolated quantum systems poses a theoretical problem that continues to cause intense discussion and research. The time-reversal-invariant Schrödinger equation furnishes a universal framework for the theoretical description of closed nonrelativistic quantum systems. How can that description be reconciled with the manifestly irreversible tendency toward statistical equilibrium encapsulated in Equation (Equation 1)? The literature on the subject is vast. For references, we confine ourselves to two review articles [5,6] summarizing a total of about 500 research papers.

The derivation of the master Equation (Equation 1) in Refs. [3,4] uses arguments that differ from the ones summarized in Refs. [5,6]. It is based upon a random-matrix approach. It is assumed that within each subsystem or class α of states, the interaction mixes the states so strongly that the resulting eigenvalues and eigenfunctions follow the statistical distribution predicted by random-matrix theory. Under that assumption, the matrix elements of the interaction connecting states in different classes α≠β become Gaussian-distributed random variables. Averaging the differential equations in time for the occupation probabilities of states in classes α=1,…,K over that distribution, and using the Markov approximation, yields master Equation (Equation 1).

In the present paper, we address two issues that have remained unexplored in Refs. [3,4]. First, we address the connection between the basic statistical assumption used in Refs. [3,4] and quantum chaos (Section 2). Second, we ask for the cause of irreversibility displayed by Equation (Equation 1). Which approximation used in the derivation causes the transition from the time-reversal-invariant Schrödinger equation to the manifestly time-reversal-noninvariant master Equation (Equation 1)? Is irreversibility due to the Markov approximation, is it due to our averaging over the statistical distribution of matrix elements of the interaction, or is there another cause? To answer that question, in Section 3, we use the results of Section 2 to define the basic random-matrix ensemble. In Section 4, we use the results of Ref. [3] to work out the time evolution of average occupation probabilities in the various classes. In Section 5, we apply the Markov approximation and introduce the central element which is responsible for the violation of the time-reversal invariance. Corrections to the Markov approximation are worked out in Section 6. Time-dependent oscillatory terms are addressed in Section 7. We summarize the results and the answers to the questions raised above in Section 8.

We focus attention on the time evolution of average occupation probabilities, i.e., on the diagonal elements of the density matrix of the system as described by Equation (Equation 1). Our derivation can be extended to the equation describing the time evolution of the full density matrix, including the off-diagonal elements. That equation is often referred to as the master equation, while Equation (Equation 1) is then called a rate equation. We do not follow that custom here. It is easily seen that upon averaging and in Markov approximation, the off-diagonal elements of the density matrix decrease exponentially in time. The decrease is governed by the same rates that determine the loss term in Equation (Equation 1).

## 2. Quantum Chaos and Random-Matrix Theory

The connection between quantum chaos and random-matrix theory (RMT) was uncovered in the late 1970s and early 1980s. In the search for “quantum signatures of classical chaos” [1], several authors turned their attention to random-matrix theory. That theory had been introduced by Wigner [7], in 1955, in an attempt to characterize the eigenvalues of the nuclear Hamiltonian that appeared as parameters in his *R*-matrix theory of nuclear reactions [8]. McDonald and Kaufman [9], Casati, Valz-Gris, and Guarneri [10], and Berry [11] investigated numerically the energy spectra of two-degrees-of-freedom quantum systems (the stadium billiard and the Sinai billiard) that are chaotic in the classical limit. Their results suggested that the distribution of spacings of neighboring eigenvalues coincides with the Wigner surmise. That surmise furnishes an excellent approximation to the distribution of nearest-neighbor spacings of eigenvalues of the Gaussian orthogonal ensemble (GOE) [12] of random matrices. Following these investigations, Bohigas, Giannoni, and Schmit [13] generated numerically a considerably larger set of eigenvalues for the Sinai billiard than had been used by the earlier authors. When combined with a refined statistical analysis, the spacings of these eigenvalues showed good agreement with the GOE fluctuation measures. The agreement caused the authors to formulate the “BGS conjecture”: The spectral fluctuation measures of a classically chaotic quantum system coincide with those of the canonical random-matrix ensemble in the same symmetry class (unitary, orthogonal, or symplectic).

The BGS conjecture has since been thoroughly tested numerically in several few-degrees-of-freedom systems (see the review in [2]). With the help of Gutzwiller’s periodic-orbit sum for the level density, Sieber and Richter [14] and Heusler, Müller, Altland, Braun, and Haake [15] gave a formal demonstration of the validity of the conjecture for general dynamical systems, summarized in the article by Müller and Sieber [16]. The demonstration is not a mathematical proof but shows physically convincingly why the BGS conjecture holds for classically chaotic quantum systems that possess a periodic-orbit sum for the level density.

These results strongly suggest that the BGS conjecture holds quite generally for strongly interacting many-body quantum systems that are chaotic in the classical limit. Ergodic theory [17] has shown that classical deterministic chaos is ubiquitous. It is characterized by positive Lyapunov exponents. These guarantee that long periodic orbits do have the properties used in Ref. [15]. However, that connection between RMT and classical chaos is too narrow in the present context. Bosonic and Fermionic systems possess symmetries that are not part of the classical ergodic theory. Ongoing research addresses the RMT fluctuation properties in such systems (see, for example, Refs. [18,19]). Moreover, there exist strongly interacting many-body systems that do not possess a well-defined classical limit. Locally interacting spin chains are an example. And yet, the spectral form factor (the Fourier transform of the spectral pair correlation function) of these systems agrees with the RMT prediction [20]. Likewise, it is not clear that very dense quantum systems like atomic nuclei do possess a classical limit. And yet, measures of spectral fluctuations evaluated for light nuclei with a Hamiltonian containing a mean-field potential and a residual two-body interaction do show agreement with RMT predictions [21]. Thermalization and random-matrix properties in many-body systems governed by a mean field and a residual interaction are analyzed more generally in Ref. [22].

It thus seems that the BGS conjecture, while generically valid, does not cover all cases of interest. In that situation, it is useful to recall the physical reason for agreement with RMT predictions in systems such as spin chains [20] or atomic nuclei [21]. It is the strong mixing of the unperturbed states of the system by the interaction. A suitable generalization of the BGS conjecture would say: For many-body systems with a sufficiently strong generic interaction, the spectral fluctuation measures agree with RMT predictions. That formulation is unspecific as it does not say how strong the interaction has to be. It is quite specific in saying that RMT spectral fluctuation properties are attained.

In conclusion, RMT spectral fluctuation properties are generic and are expected to occur universally in strongly interacting many-body systems. Therefore, the assumption formulated in Section 1 is on firm grounds. Nevertheless, for a specific system, it is advisable to make sure that RMT spectral fluctuation properties prevail, either by demonstrating that the system is classically chaotic or by a numerical test. Attention must be paid, in particular, to one-body and many-body localization [23,24,25]. Either phenomenon is not compatible with RMT spectral fluctuation properties.

## 3. Ensemble of Random Matrices

Before we address the second issue raised in the Introduction, we give a brief summary of the assumptions and developments in Ref. [4]. We define the basic random-matrix ensemble.

It is assumed that in Hilbert space, the states of the system can be grouped into a finite number *K* of classes, such that within each class, the states interact strongly. The example considered in Ref. [4] is a system of Fermions with a mean-field average potential and a closed-shell ground state. Excited states are defined as particle-hole excitations out of the ground state. Classes of such states carry the same number of particle-hole states. Our theoretical development is more general, however, and is not restricted to that example. We confine ourselves, however, to the case of orthogonal symmetry.

In every class α, we use a basis of orthonormal states labeled |αm〉 with $m=1,2,\ldots ,N_{\alpha }$ and $N_{\alpha }\gg 1$. On that basis, the Hamiltonian is written as
(3)$$\begin{array}{c} H_{\alpha m,\beta n}=\delta_{\alpha \beta }H_{mn}^{\left(\alpha \right)}+\langle \alpha m|V|\beta n\rangle . \end{array}$$
In Ref. [4], it is assumed that within each class α, the Hamiltonian $H^{\left(\alpha \right)}$ is a member of the GOE. That is in line with the arguments of Section 2. The non-statistical real matrix elements 〈αm|V|βn〉 connect different classes and vanish for α=β. They also vanish unless the states in both classes carry the same conserved quantum numbers. In Ref. [4], it was assumed that *V* is a two-body interaction. That interaction changes the particle-hole number by one unit. It does not connect classes of particle-hole states differing in particle-hole number by two or more units. In the present context, it is likewise admissible that *V* does not connect every class with every other one. For definiteness, we exclude, however, the possibility that the classes form subgroups that are not connected to each other at all by matrix elements of *V*. The mixing of the states in different classes due to *V* yields eigenfunctions that are spread over several or all classes. As we shall see, that fact ultimately causes statistical equilibration as described by Equation (Equation 1).

The diagonalization of each of the GOE Hamiltonians $H^{\left(\alpha \right)}$ in Equation (Equation 3) yields for the total Hamiltonian in Equation (Equation 3) the expression
(4)$$\begin{array}{c} H_{\alpha \mu ,\beta \nu }=E_{\alpha \mu }\delta_{\alpha \beta }\delta_{\mu \nu }+\langle \alpha \mu |V|\beta \nu \rangle . \end{array}$$
The eigenvalues $E_{\alpha \mu }$ follow Wigner–Dyson statistics. The projections of the eigenfunctions |αμ〉 onto some fixed vector |αm〉 have a Gaussian distribution. It follows that the matrix elements 〈αμ|V|βν〉 have a Gaussian distribution with respect to their dependence on both μ and ν.

The central step taken in Ref. [4] consists of replacing in each class α the Hamiltonian $H^{\left(\alpha \right)}$ by an ensemble of GOE Hamiltonians. That implies that the $E_{\alpha \mu }$ in Equation (Equation 4) form an ensemble of Wigner–Dyson-distributed eigenvalues and that the matrix elements 〈αμ|V|βν〉 form an ensemble of Gaussian-distributed zero-centered random variables with second moments
(5)$$\begin{array}{ccc} & & \langle \langle \alpha \mu |V|\beta \nu \rangle \langle \alpha^{\prime }\mu^{\prime }\left|V\right|\beta^{\prime }\nu^{\prime }\rangle \rangle \\ & & =V_{\alpha \beta }^{2}(\delta_{\alpha \alpha^{\prime }}\delta_{\mu \mu^{\prime }}\delta_{\beta \beta^{\prime }}\delta_{\nu \nu^{\prime }}+\delta_{\alpha \beta^{\prime }}\delta_{\mu \nu^{\prime }}\delta_{\beta \alpha^{\prime }}\delta_{\nu \mu^{\prime }}). \end{array}$$
The big angular brackets denote the ensemble average. For α≠β, the states |αν〉 and |βν〉 are statistically uncorrelated. That gives rise to the Kronecker deltas involving class labels in Equation (Equation 5). The mean square matrix elements $V_{\alpha \beta }^{2}$ vanish for α=β. They measure the strength of the coupling of classes (α,β). Within each class α, we eventually take the GOE limit of infinite matrix dimension $N_{\alpha }$. Then, the spectrum of $H^{\left(\alpha \right)}$ extends from −2λ to +2λ, the average level spacing $d_{\alpha }\propto 1/N_{\alpha }$ tends to zero, and the average level density $\rho_{E,\alpha }$ versus energy *E* is given by
(6)$$\begin{array}{c} \rho_{E,\alpha }=\frac{N}{\pi \lambda }\sqrt{1-{\left(E/\left(2\lambda \right)\right)}^{2}}. \end{array}$$

In the following sections, we display the assumptions under which Equation (Equation 1) follows from that central step. We show that Equation (Equation 1) describes the time evolution of ensemble-averaged occupation probabilities defined for each class. Therefore, statistical equilibration does not necessarily occur for each member of the ensemble. It is an average property, characteristic of most members of the ensemble.

## 4. Time Evolution of Average Occupation Probabilities

To implement the approach, we use the time-dependent Schrödinger equation in the interaction representation. The wave function is expanded on the basis of states |αμ〉,|βν〉,… with time-dependent occupation amplitudes $c_{\alpha \mu }\left(t\right),c_{\beta \nu }\left(t\right),\ldots$. That yields
(7)$$\begin{array}{c} i\hbar {\dot{c}}_{\alpha \mu }\left(t\right)=\sum_{\beta \nu }\langle \alpha \mu |\tilde{V}\left(t\right)|\beta \nu \rangle c_{\beta \nu }\left(t\right). \end{array}$$
The dot indicates the time derivative, and the time-dependent matrix element
(8)$$\begin{array}{c} \langle \alpha \mu |\tilde{V}\left(t\right)|\beta \nu \rangle =\langle \alpha \mu |V|\beta \nu \rangle \exp\left\{i\left(E_{\alpha \mu }-E_{\beta \nu }\right)t/\hbar \right\} \end{array}$$
is Hermitean, $\langle \alpha \mu |\tilde{V}{\left(t\right)|\beta \nu \rangle }^{*}=\langle \beta \nu |\tilde{V}\left(t\right)|\alpha \mu \rangle$.

Time-dependent average occupation probabilities $P_{E,\alpha }\left(t\right)$ of the states in class α are defined as
(9)$$\begin{array}{c} P_{E,\alpha }\left(t\right)=\langle \sum_{\mu }{\left|c_{\alpha \mu }\left(t\right)\right|}^{2}\rangle . \end{array}$$
In the GOE limit, the sum over μ encompasses states |αμ〉 with energies $E_{\alpha \mu }$ located within a very small energy interval centered on the energy *E* defined by the initial condition $c_{\alpha \mu }\left(0\right)=\delta_{\alpha \alpha_{0}}\delta_{\mu \mu_{0}}$ at time t=0 for the coefficients $c_{\alpha \mu }\left(t\right)$. The time derivative of $P_{E,\alpha }\left(t\right)$ is
(10)$$\begin{array}{ccc} {\dot{P}}_{E,\alpha }\left(t\right) & = & \langle \sum_{\mu }\left(c_{\alpha \mu }^{*}{\dot{c}}_{\alpha \mu }+{\dot{c}}_{\alpha \mu }^{*}c_{\alpha \mu }\right)\rangle . \end{array}$$
In Ref. [4], the average in Equation (Equation 10) is calculated by expanding each of the factors $c_{\alpha \mu }\left(t\right)$, $c_{\alpha \mu }^{*}\left(t\right)$, ${\dot{c}}_{\alpha \mu }\left(t\right)$, ${\dot{c}}_{\alpha \mu }^{*}\left(t\right)$ on the right-hand side perturbatively in powers of $\tilde{V}$. That results in a multiple sum over products of Gaussian-distributed matrix elements. The average is calculated for each such product separately. In each product, the average equals the sum over all ways of averaging pairs of matrix elements using Equation (Equation 5). The number of such pairwise “contractions” of matrix elements proliferates with increasing order of the perturbation expansion. Terms of leading order are obtained by suppressing all contributions that contain derivatives of the average level density $\rho_{E,\alpha }$ with respect to energy *E* in any class α. Such suppression allows resummation of the series, is tantamount to the Markov approximation, and yields Equation (Equation 1) for the time evolution of $P_{E,\alpha }\left(t\right)$. Leading-order contraction patterns are displayed in Section 5 of Ref. [3].

Here, we present an alternative derivation of Equation (Equation 1) which is simpler, more transparent, and more general than the one used in Ref. [4]. That allows us to answer the second question raised in Section 1.

We consider the first term on the right-hand side of Equation (Equation 10). The contraction rules given in Ref. [3] imply that the first factor *V* in the perturbative expansion of ${\dot{c}}_{\alpha \mu }$ is contracted either with the factor *V* immediately following it (case (i)), or with the first such factor in the perturbative expansion of $c_{\alpha \mu }^{*}$ (case (ii)). For case (i), we integrate Equation (Equation 7) with respect to time and reinsert the result on the right-hand side of that equation. That gives
(11)$$\begin{array}{ccc} & & {\dot{c}}_{\alpha \mu }\left(t\right)=\frac{1}{i\hbar }\sum_{\beta \nu }\langle \alpha \mu |\tilde{V}\left(t\right)|\beta \nu \rangle c_{\beta \nu }\left(0\right)\\ & & +\frac{1}{{\left(i\hbar \right)}^{2}}\sum_{\beta \nu }\langle \alpha \mu |\tilde{V}\left(t\right)|\beta \nu \rangle \\ & & \;\times \int_{0}^{t}dt_{1}\sum_{\gamma \rho }\langle \beta \nu |\tilde{V}\left(t_{1}\right)|\gamma \rho \rangle c_{\gamma \rho }\left(t_{1}\right). \end{array}$$
The first term on the right-hand side does not contribute because $\tilde{V}$ is not followed by a second such factor. We insert the second term into $\sum_{\mu }\langle c_{\alpha \mu }^{*}{\dot{c}}_{\alpha \mu }\rangle$ and contract the two factors *V* displayed explicitly. We obtain
(12)$$\begin{array}{ccc} & & \frac{1}{{\left(i\hbar \right)}^{2}}\sum_{\beta }V_{\alpha \beta }^{2}\sum_{\mu \nu }\int_{0}^{t}dt_{1}\exp\left\{i\left(E_{\alpha \mu }-E_{\beta \nu }\right)\left(t-t_{1}\right)/\hbar \right\}\\ & & \;\times \langle c_{\alpha \mu }^{*}\left(t\right)c_{\alpha \mu }\left(t_{1}\right)\rangle . \end{array}$$
Writing $t_{1}=t+\left(t_{1}-t\right)$, we expand $\langle c_{\alpha \mu }\left(t_{1}\right)\rangle$ in a Taylor series in $\left(t_{1}-t\right)$ and keep the first two terms. The leading-order term yields the Markov approximation, and the next-order term, the correction of subleading order. We define $\Delta_{\nu }=\left(E_{\alpha \mu }-E_{\beta \nu }\right)/\hbar$, carry out the time integrations, and find
(13)$$\begin{array}{ccc} & & \frac{1}{{\left(i\hbar \right)}^{2}}\sum_{\beta }V_{\alpha \beta }^{2}\sum_{\mu \nu }\{\frac{i}{\Delta_{\nu }}(1-\exp\{i\Delta_{\nu }t\})\langle c_{\alpha \mu }^{*}\left(t\right)c_{\alpha \mu }\left(t\right)\rangle \\ & & \;+\frac{1}{\Delta_{\nu }^{2}}(1+\left(it\Delta_{\nu }-1\right)\exp\left\{i\Delta_{\nu }t\right\})\langle c_{\alpha \mu }^{*}\left(t\right){\dot{c}}_{\alpha \mu }\left(t\right)\rangle \}. \end{array}$$
For case (ii) we proceed analogously. In the first term, on the right-hand side of Equation (Equation 10), we use Equation (Equation 7) for ${\dot{c}}_{\alpha \mu }$. For $c_{\alpha \mu }^{*}$, we integrate the complex conjugate of Equation (Equation 7) over time and obtain
(14)$$\begin{array}{c} c_{\alpha \mu }^{*}\left(t\right)=\frac{i}{\hbar }\sum_{\beta \nu }\int_{0}^{t}dt_{1}\;\langle \alpha \mu |\tilde{V}\left(t_{1}\right){|\beta \nu \rangle }^{*}c_{\beta \nu }^{*}\left(t_{1}\right). \end{array}$$
The term analogous to expression (Equation 12) is
(15)$$\begin{array}{ccc} & & \frac{1}{{\left(\hbar \right)}^{2}}\sum_{\beta }V_{\alpha \beta }^{2}\sum_{\mu \nu }\int_{0}^{t}dt_{1}\exp\left\{i\left(E_{\alpha \mu }-E_{\beta \nu }\right)\left(t-t_{1}\right)/\hbar \right\}\\ & & \;\times \langle c_{\beta \nu }^{*}\left(t_{1}\right)c_{\beta \nu }\left(t\right)\rangle . \end{array}$$
We expand $c_{\beta \nu }^{*}\left(t_{1}\right)$ in a Taylor series in $t_{1}-t$ and keep the first two terms. We define $\Delta_{\mu }=E_{\alpha \mu }-E_{\beta \nu }/\hbar$, carry out the time integrations, and obtain
(16)$$\begin{array}{ccc} & & \frac{1}{{\left(\hbar \right)}^{2}}\sum_{\beta }V_{\alpha \beta }^{2}\sum_{\mu \nu }\{\frac{i}{\Delta_{\mu }}(1-\exp\{i\Delta_{\mu }t\})\langle c_{\beta \nu }^{*}\left(t\right)c_{\beta \nu }\left(t\right)\rangle \\ & & \;+\frac{1}{\Delta_{\mu }^{2}}(1+\left(it\Delta_{\mu }-1\right)\exp\left\{i\Delta_{\mu }t\right\})\langle {\dot{c}}_{\beta \nu }^{*}\left(t\right)c_{\beta \nu }\left(t\right)\rangle \}. \end{array}$$

In evaluating expressions (Equation 13) and (Equation 16), we take the GOE limit defined on top of Equation (Equation 6). We omit the terms carrying the factor $\exp\{i\Delta_{\nu }t\}$ in expression (Equation 13) and the terms carrying the factor $\exp\{i\Delta_{\mu }t\}$ in expression (Equation 16). These terms are addressed in Section 7.

## 5. Markov Approximation

We focus attention on the remaining terms in the first lines of expressions (Equation 13) and (Equation 16). These are given by
(17)$$\begin{array}{ccc} & & -\frac{1}{\hbar^{2}}\sum_{\beta }V_{\alpha \beta }^{2}\sum_{\mu \nu }\left(\frac{i}{\Delta_{\nu }}\langle c_{\alpha \mu }^{*}\left(t\right)c_{\alpha \mu }\left(t\right)\rangle +c.c.\right)\;,\\ & & \;\;\frac{1}{\hbar^{2}}\sum_{\beta }V_{\alpha \beta }^{2}\sum_{\mu \nu }\left(\frac{i}{\Delta_{\mu }}\langle c_{\beta \nu }^{*}\left(t\right)c_{\beta \nu }\left(t\right)\rangle +c.c.\right). \end{array}$$
In the summation over ν in the first line, we assume that $\Delta_{\nu }$ carries an infinitesimal positive imaginary increment iε with ε>0. In the GOE limit, that gives
(18)$$\begin{array}{c} \frac{1}{\Delta_{\nu }}=P\frac{\hbar }{E_{\alpha \mu }-E_{\beta \nu }}-i\pi \hbar \delta \left(E_{\alpha \mu }-E_{\beta \nu }\right). \end{array}$$
The summation over ν is replaced by an integral over $E^{\prime }=E_{\beta \nu }$. The weight factor is $\rho_{E^{\prime },\beta }$, the average level density in class β defined in Equation (Equation 6). The symbol P denotes the principal-value integral. In the first expression (Equation 17), that integral gives a purely imaginary contribution which cancels against the principal-value integral in the complex conjugate term. We are left with the contribution due to the delta function which links only states in classes α and β that pertain to the same energy. As a consequence, the process is entirely on shell. The integration over $E^{\prime }$ yields $-i\pi \hbar \rho_{E_{\alpha \mu },\beta }$. We recall that the summation over μ encompasses a set of energies $E_{\alpha \mu }$ narrowly centered at *E*, the energy defined by the initial value for $c_{\alpha \mu }$. Within that interval, we may replace $\rho_{E_{\alpha \mu },\beta }$ with $\rho_{E,\beta }$. We use definition (Equation 9) to perform the remaining summation over μ and obtain for the first of expressions (Equation 17)
(19)$$\begin{array}{c} -\frac{2\pi }{\hbar }P_{E,\alpha }\left(t\right)\sum_{\beta }V_{\alpha \beta }^{2}\rho_{E,\beta }. \end{array}$$
That is the loss term in Equation (Equation 1), with an explicit expression for the rate $R_{\alpha \to \beta }$. Proceeding analogously for the second term in expression (Equation 17), we obtain
(20)$$\begin{array}{c} +\frac{2\pi }{\hbar }\rho_{E,\alpha }\sum_{\beta }V_{\alpha \beta }^{2}P_{E,\beta }\left(t\right). \end{array}$$
That is the gain term in Equation (Equation 1), again with an explicit expression for the rate $R_{\beta \to \alpha }$. Our treatment has been very explicit because gain term plus loss term together yield Equation (Equation 1) which violates time-reversal invariance. The origin of that violation and of the resulting tendency toward equilibration is in the choice of the sign of the infinitesimal increment iε in $\Delta_{\nu }$ and $\Delta_{\mu }$. That choice determines the sign of the last term in Equation (Equation 18). Choosing the opposite sign would change the signs of gain term and loss term and would result in an equation where the occupation probabilities $P_{E,\alpha }$ grow indefinitely with time. That is physically unacceptable.

Use of an imaginary increment iε in $\Delta_{\nu }$ and in $\Delta_{\mu }$ is a formal necessity. Without it, our approach would involve the matrix elements $\langle \alpha m|{\left(E-H^{\left(\alpha \right)}\right)}^{-1}|\alpha m^{\prime }\rangle$ of the propagators of the GOE Hamiltonians $H^{\left(\alpha \right)}$ with real energy *E* for α=1,…,K. Each of these quantities is a random variable that does not possess a well-defined distribution. All moments higher than the first diverge. That can easily be checked by direct calculation. Use of an imaginary increment iε in $\Delta_{\nu }$ and in $\Delta_{\mu }$ is also a physical necessity. It breaks time-reversal invariance. Without it, the terms in expressions (Equation 19) and (Equation 20) would vanish, and the Markov approximation would say that the time derivatives of all average occupation probabilities are equal to zero. We see that a consistent and singularity-free derivation of Equation (Equation 1) cannot be given in the present framework for ε=0. These difficulties are removed by adding the increment iε. Purely formally, either sign of ε is admissible. Choosing a positive value for ε reflects physical necessity and has distinct physical significance. The sign of ε determines the signs of gain and loss term and, thereby, the direction of the arrow of time. A positive value of ε causes an ever-so-small exponential decrease with time of all time-dependent exponentials in the time evolution of $c_{\alpha \mu }\left(t\right)$ and $c_{\alpha \mu }^{*}\left(t\right)$. The resulting small but continuous loss of occupation probability corresponds to a leakage of the system, physically caused by the evaporation of particles, by gamma emission, or by another such mechanism. That leakage determines physically the direction of time and is ultimately responsible for the loss of time-reversal invariance in Equation (Equation 1).

In summary: In the present framework, master Equation (Equation 1) cannot be derived for a strictly isolated system, i.e., for ε=0. The direction of the arrow of time is determined by the sign of ε. A positive value of ε accounts for an (ever-so-small) coupling to the outside world that causes a loss of probability.

## 6. Corrections to the Markov Approximation

Subleading-order corrections to the Markov approximation are due to the terms in the second lines of expressions (Equation 13) and (Equation 16) and to certain contraction patterns of pairs of matrix elements of *V* that violate the rules laid down in Ref. [3] to leading order. We consider these in turn.

In the second lines of expressions (Equation 13) and (Equation 16), we neglect the strongly oscillating terms. The remaining expressions are proportional, respectively, to $\sum_{\nu }1/\Delta_{\nu }^{2}$ and to $\sum_{\mu }1/\Delta_{\mu }^{2}$. Either term can be written as a derivative with respect to energy. It follows that the said terms are proportional, respectively, to ${\dot{P}}_{E,\alpha }\sum_{\beta }V_{\alpha \beta }^{2}d\rho_{E,\beta }/dE$ and to $d\rho_{E,\alpha }/dE\sum_{\beta }V_{\alpha \beta }^{2}{\dot{P}}_{E,\beta }$. That result has two consequences. First, it contains the time derivatives of the occupation probabilities $P_{E,\alpha }$ and $P_{E,\beta }$. When statistical equilibrium is reached, all time derivatives vanish. The equilibrium solution $P_{E,\alpha }^{\left(0\right)}=C\rho_{E,\alpha }$ is determined by the interplay of gain term and loss term on the right-hand side of Equation (Equation 1). It has the same value without and with non-Markovian corrections. We conclude that the corrections to the Markov approximation in expressions (Equation 13) and (Equation 16) change the form of master Equation (Equation 1), may affect the time within which statistical equilibrium is approximately attained, but do not affect the tendency of the system to attain equilibrium or change the form of the equilibrium solution. Second, the result allows us to sharpen the criterion for the validity of the Markov approximation. The corrections to the Markov approximation are negligible if the characteristic energy interval $E_{\beta }=\rho_{E,\beta }/\left(d\rho_{E,\beta }/dE\right)$, within which the level density $\rho_{E,\beta }$ changes significantly, is large compared to the spreading width $2\pi V_{\alpha \beta }^{2}\rho_{E,\beta }$ for all classes α coupled to class β.

We turn to the subleading contributions due to contraction patterns that violate the rules laid down in Ref. [3]. We confine ourselves to a single example. In Equation (Equation 11), we carry the perturbation expansion of ${\dot{c}}_{\alpha \mu }\left(t\right)$ up to the term of fourth order in $\tilde{V}$. That term involves a time-ordered integral over times $t_{1},t_{2},t_{3}$. The integrand is the product of three matrix elements of $\tilde{V}$ and of $c_{\gamma \rho }\left(t_{3}\right)$. In the average over the four matrix elements of *V* displayed explicitly, the contribution of subleading order is obtained by contracting the outer pair and the inner pair. In the resulting expression, we replace $c_{\alpha \mu }\left(t_{3}\right)$ by $c_{\alpha \mu }\left(t\right)$ and carry out the three time integrations. As done at the end of Section 4, we suppress terms that carry rapidly oscillating exponential factors. These are addressed in the next section. That leaves us with a single term given by
(21)$$\begin{array}{ccc} & & \frac{1}{{\left(i\hbar \right)}^{4}}\sum_{\beta \gamma }V_{\alpha \beta }^{2}V_{\beta \gamma }^{2}\sum_{\mu \nu }\frac{\hbar^{2}}{i^{2}{\left(E_{\alpha \mu }-E_{\beta \nu }\right)}^{2}}\\ & & \;\times \sum_{\rho }\frac{\hbar }{i(E_{\gamma \rho }-E_{\alpha \mu })}\langle c_{\alpha \mu }^{*}\left(t\right)c_{\alpha \mu }\left(t\right)\rangle . \end{array}$$
The sum over ν can be written as a derivative with respect to energy. We use Equation (Equation 18) and suppress the principal-value contributions. The result is proportional to $iP_{E,\alpha }\left(t\right)\sum_{\beta }V_{\alpha \beta }^{2}d\rho_{E,\beta }/dE\sum_{\gamma }V_{\beta \gamma }^{2}\rho_{E,\gamma }$. It is purely imaginary and, thus, does not contribute to Equation (Equation 1). We note, however, that the form of the result shows that the criterion for the validity of the Markov approximation stated in the previous paragraph applies here as well. A term similar to expression (Equation 21) is obtained from the subleading contraction connecting a factor *V* in ${\dot{c}}_{\alpha \mu }\left(t\right)$ and three factors *V* in $c_{\alpha \mu }^{*}\left(t\right)$. Actually, neither of these two terms modifies Equation (Equation 1). The form of the equilibrium solution remains unchanged.

These results confirm that the Markov approximation is not the cause of violation of time-reversal invariance in Equation (Equation 1).

## 7. Oscillatory Terms

As done in Ref. [4], we have consistently omitted terms of the form
(22)$$\begin{array}{c} \sum_{\nu }\frac{\exp i(E-E_{\beta \nu })t/\hbar \}}{E-E_{\beta \nu }+i\varepsilon }. \end{array}$$
Such terms are not peculiar to the Markov approximation. They arise, in general, whenever the time evolution of a quantum system is studied perturbatively. Actually, the contents of the big round brackets in Equations (Equation 13) and (Equation 16) vanish for t=0. That shows that the neglect of the oscillatory terms, intuitively justified because of the rapid oscillation of the exponential, is valid only beyond some finite time $t_{0}$. At that time, the contribution of the oscillating terms becomes small compared to that of the first term in big round brackets in Equations (Equation 13) and (Equation 16). To determine $t_{0}$, we replace the infinitesimal increment ε in expression (Equation 22) by a finite width γ. That gives
(23)$$\begin{array}{c} \sum_{\nu }\frac{\exp i(E-E_{\beta \nu })t/\hbar \}}{{\left(E-E_{\beta \nu }\right)}^{2}+\gamma^{2}}\left(E-E_{\beta \nu }-i\gamma \right)\;. \end{array}$$
Upon summation, the real part of the big round bracket vanishes or is very small. That is true even for t=0. The term proportional to iγ involves the summation over a Lorentzian with width γ. If the remaining terms in the sum were sufficiently smooth functions of $E_{\beta \nu }$, the limit γ→0 may be taken. That yields the delta function on the right-hand side of Equation (Equation 18). In the present case, we cannot use that step because of the oscillation of the exponential. We assume that $\gamma \gg d_{\beta }$, the mean level spacing in class β. Then, the sum over the Lorentzian effectively involves a large number of terms. Because of the exponential, the sum becomes negligible for t≳2πħ/γ. We compare that with the characteristic time scale of Equation (Equation 1) which is given by the inverse rate $\hbar /(2\pi V_{\alpha \beta }^{2}\rho_{E,\beta })$. The two time scales are approximately equal, and oscillating terms may be suppressed in master Equation (Equation 1) after some initial time $t_{0}=2\pi \hbar /\gamma$, if γ is of order ${\left(2\pi \right)}^{2}V_{\alpha \beta }^{2}\rho_{E,\beta }$.

While an infinitesimal increment ε is sufficient to guarantee violation of time-reversal invariance in Equation (Equation 1), a finite width γ is needed to validate omission of the exponential terms for times $t≳t_{0}=2\pi \hbar /\gamma$. The width γ, caused by decay into open channels, leads to a decay in time of all exponentials in the time evolution of amplitudes, see Equation (Equation 7). To be consistent, such decay may have to be accounted for explicitly in terms of an additional loss term in Equation (Equation 1). That was done, for instance, in Ref. [26], where γ accounts for neutron evaporation in a laser-induced nuclear reaction.

## 8. Summary and Conclusions

We have considered a system composed of classes of states, each of which is strongly mixed internally. By recalling central results of quantum chaos and its connection to random-matrix theory, we have justified the use of a random-matrix approach to that system. Within each class, the dynamics are governed by an ensemble of GOE Hamiltonians. States in different classes are coupled by an ensemble of Gaussian-distributed matrix elements. We have simplified the derivation of Ref. [4] for the time evolution of average occupation probabilities for each class. That has enabled us to precisely define the conditions of validity of master Equation (Equation 1) and to identify the central element that causes the violation of time-reversal invariance. Irreversibility is due to an ever-so-small loss of probability (due to particle evaporation, gamma emission, or another such mechanism). The loss term defines the direction of the arrow of time. In combination with ensemble averaging and the Markov approximation, it leads to the irreversible master Equation (Equation 1).

Corrections of the subleading order to the Markov approximation may modify the speed at which the system equilibrates. They do not affect the irreversible tendency toward statistical equilibrium. The corrections are negligible, and the Markov approximation is valid, if in each class α the average level density $\rho_{E,\alpha }$ is sufficiently smooth. Quantitatively, that requires that the energy interval $E_{\alpha }=\rho_{E,\alpha }/\left(d\rho_{E,\alpha }/dE\right)$, within which $\rho_{E,\alpha }$ changes significantly, is large in comparison with every one of the spreading widths $2\pi V_{\beta \alpha }^{2}\rho_{E,\alpha }$ that feed states in class α from states in any other class β. For the terms that strongly oscillate in time, we have derived a lower bound $t_{0}$ on time beyond which such oscillating terms are negligible.

We have answered the questions raised in Section 1. In doing so, we have formulated precise conditions on level densities and time scales within which Equation (Equation 1) can be used with confidence.

After completion of the manuscript, I became aware of Ref. [27]. I am grateful to A. Volya and to V. Zelevinsky for drawing my attention to that paper. Investigating a number of examples, the authors demonstrate the limitations in specific cases of the universal random-matrix approach adopted in the present paper.

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
