# Peer review of "Quantum Chaos, Random Matrices, and Irreversibility in Interacting Many-Body Quantum Systems"

_entropy, 2022, doi:10.3390/e24070959_

Round 1
Reviewer 1 Report
The manuscript addresses an interesting problem of RMT-like description of a closed systems as composed of several parts and provides a derivation of Pauli master equations (1). This set of equations considers, however, probabilities only, i.e. diagonal parts of the density matrix. The full master equation (typically developed for two interacting subsystems) involves also off diagonal elements, the so called coherences. I believe it would be profitable for the reader if the author discusses which approximations allow him to neglect, in the model studied, the corresponding terms. I would rather call (1) as a set of rate equations and not master equations.
The weakness of the present paper is that it relies heavily on the content of [3,4].
The single reference to MBL review (Alet) was published, I believe in 2018.
I suggest to add other reviews to this rapidly developing field, e.g. Abanin et al. 2019 RMP.
Author Response
Reply to Referee 1:
I am grateful to the refereee for his comments and suggestions.
In answer to his remarks I have added a paragraph at the end of the Introduction. I hope that
clarifies the issue.
My reliance on Refs. [3,4] is caused by the fact that I am not aware of other works using random-
matrix theory to derive the master equation.
I have added the reference to Abanin et al.

Reviewer 2 Report
Based on the authors' previous works [3,4] and triggered by an ongoing discussion of equilibration in isolated quantum systems, Prof. Weidenmueller aims to clarify a connection between quantum chaos, its random matrix theory description and their role in justification of the Pauli master equation which describes statistical equilibration in closed quantum systems.
Results of the author's analysis (which is sharp, clear and convincing) are summarized in the closing section of the paper.
I suggest that the paper be accepted in its present form.
Author Response
Reply to Referee 2:
I am grateful to the referee for his comments.

Reviewer 3 Report
Thank you for sending me the article by Prof. Weidenmueller. It is an interesting and well-timed study in the area of current interest for many branches of modern physics, from nuclei to condensed matter and quantum informatics. It is also absolutely appropriate for the volume devoted to Giulio Casati. From the philosophical viewpoint it is important to stress as it is done here in the Introduction that it is not appropriate in general to talk about quantum chaos just as a remnant of classical chaotic dynamics, the actual situation might be rather opposite.
I allow you to make my message open to Prof. Weidenmueller. Not long ago, we published in a new journal an article on a related subject, that possibly was not noticed by Prof. Weidenmueller:
A. Volya and V. Zelevinsky,
Time-dependent relaxation of observables in complex quantum systems.
J. Phys. Complexity 1, 025007 (2020); arXiv:1905.11918.
It seems that our approach is not identical to that in the article under review and the conclusions are not fully equivalent as well. Therefore I would be very glad to know the Author's opinion with the corresponding reference that would give the readers a chance to get more food for making their own opinion on this topic.
Author Response
Reply to Zelevinsky:
Dear Vladimir,
I thank you for drawing my attention to your paper. In response I have added a paragraph at the
very end of the paper which I believe reflects what you said (to which I fully agree) plus a reference
to your paper. Best regards Hans
